# Hydrological Model Performance in the Verde River Basin, Minas Gerais, Brazil

Conceição de M. M. de Oliveira [1,2], Lívia A. Alvarenga [1,*], Samuel Beskow [3], Zandra Almeida da Cunha [3], Marcelle Martins Vargas [3], Pâmela A. Melo [1], Javier Tomasella [4], Ana Carolina N. Santos [4], Vinicius S. O. Carvalho [5] and Vinicius Oliveira Silva [1]

[1] Department of Water Resources, Federal University of Lavras, Lavras 37200-000, Brazil; cita_mo@yahoo.com.br (C.d.M.M.d.O.); pamela.ap.melo@gmail.com (P.A.M.); vosengambiental@gmail.com (V.O.S.)

[2] Department of Agricultural Engineering, State University of Maranhão, Cidade Universitária Paulo VI., São Luis 65055-310, Brazil

[3] Center for Technological Development/Water Resources Engineering, Federal University of Pelotas, Pelotas 96010-610, Brazil; samuelbeskow@gmail.com (S.B.); zandraacunha@gmail.com (Z.A.d.C.); marcellevarg@gmail.com (M.M.V.)

[4] National Institute for Space Research (INPE), National Center for Monitoring and Early Warning of Natural Disasters (CEMADEN), Postal Code 01, Cachoeira Paulista 12630-970, Brazil; javier.tomasella@inpe.br (J.T.); anacarolnsantos@gmail.com (A.C.N.S.)

[5] Institute of Natural Resources, Federal University of Itajubá, Itajubá 37500-903, Brazil; vsiqueira18@gmail.com

\* Correspondence: livia.aalvarenga@ufla.br

**Abstract:** In hydrological modelling, it is important to consider the uncertainties related to a model's structures and parameters when different hydrological models are used to represent a system. Therefore, an adequate analysis of daily discharge forecasts that takes into account the performance of hydrological models can assist in identifying the best extreme discharge forecasts. In this context, this study aims to evaluate the performance of three hydrological models—Lavras Simulation of Hydrology (LASH), Variable Infiltration Capacity (VIC), and Distributed Hydrological Model (MHD-INPE) in the Verde River basin. The results demonstrate that LASH and MHD can accurately simulate discharges, thereby establishing them as crucial tools for managing water resources in the study region's basins. Moreover, these findings could serve as a cornerstone for future studies focusing on food and water security, particularly when examining their connection to climate change scenarios.

**Keywords:** LASH; VIC; MHD-INPE; performance; hydrological models





## 1. Introduction

Hydrological models have proven to be effective instruments for enhancing the comprehension of hydrological phenomena in basins [1]. These models have versatile applications, including discharge prediction, the evaluation of water availability, and the investigation of the impacts of climate and land-use modifications [2–5].

The selection of a hydrological model should be based on the intended application since each model has its own assumptions and limitations. The use of conceptual and distributed hydrological models is justified by their ability to satisfactorily represent the physical processes occurring within a basin while accounting for the spatial variability of physical and meteorological parameters [6,7]. Therefore, factors such as the availability of data in a region, the number of required parameters, and the level of description of hydrological processes must all be taken into consideration.

The Soil and Water Assessment Tool (SWAT) [8,9] and the Variable Infiltration Capacity (VIC) models [10–12] are semi-distributed models utilized in hydrological simulations of the Verde River basin in Minas Gerais state, Brazil [5,7]. These models have produced satisfactory results at a monthly time step, suggesting the possibility of conducting further

research to assess the impacts of climate and land-cover changes [7]. However, it is important to acknowledge that these studies simplify some numerical processes, as their objective is to evaluate hydrological impacts in average terms. In contrast, the estimation of daily discharge is a more dynamic and complex process capable of representing extreme discharge events and periods of drought on a daily basis. Therefore, ensuring good performance of hydrological models at a daily time step is essential.

The Lavras Simulation of Hydrology (LASH) model has undergone testing in various basins located in different states of Brazil, including Minas Gerais, Rio Grande do Sul, Amazonas, and Tocantins, representing diverse biomes, scales, and edaphoclimatic conditions. For instance, Viola et al. [13], Mello et al. [14], and Beskow et al. [15] have conducted research using LASH concerning the headwaters of the Rio Grande basin. While various applications of LASH can be highlighted, no study has yet been published comparing the performance of different hydrological models (LASH, VIC, and MHD-INPE) at a daily time step. Such a comparison could yield a more robust analysis of peak and recession discharge magnitudes.

In addition to the hydrological models mentioned (SWAT, VIC, and LASH), the Distributed Hydrological Model (MHD-INPE) is also a viable option for discharge forecasting. The MHD-INPE [16] is a distributed model that was developed at the Instituto Nacional de Pesquisas Espaciais (INPE), and it is an adaptation of the Large Basin Model that was developed at the Instituto de Pesquisas Hidráulicas (MGB-IPH). The MHD-INPE has been successfully applied to basins of varying sizes, ranging from 5 to 1.4 million km$^2$, for studies on land-use and land-cover changes, climate change and hydropower, discharge forecasting, and water sustainability, for which it has consistently yielded satisfactory results [17–21].

The Rio Verde watershed, covering an area of 4100 km$^2$, is located in the southern region of Minas Gerais, situated within the Atlantic Forest biome in the Serra da Mantiqueira. This basin forms part of the Serra da Mantiqueira Mountain range, which plays a vital role in biodiversity conservation within the Atlantic Forest biome. The Serra da Mantiqueira region encompasses several headwater regions that contribute to significant rivers in terms of water supply, irrigation, and hydropower generation. Moreover, this area offers considerable economic importance with respect to agricultural development, especially regarding coffee production, in the southern part of the state of Minas Gerais.

Hydrological models may exhibit similar performance, but the average discharge response can vary. Therefore, studies utilizing different hydrological models can uncover potential sources of uncertainty in discharge forecasting arising from variations in calibration methodologies, model structures, process equations, and input data availability, among other factors. The novelty of this study lies in its assessment of the performance of three hydrological models (LASH, VIC, and MHD) at a daily time step in the Verde River Basin, a region of economic and biodiversity significance in Minas Gerais State, Brazil. The study aimed to address three key questions: (i) Is there any difference in the performance of the models? (ii) Can the models adequately simulate peak and low discharges in the Verde River Basin? (iii) Are the models suitable for water resources management in the Verde River Basin? These questions seek to uncover the performance of hydrological models in the South of the state of Minas Gerais and their potential application across the region.

## 2. Material and Methods

### 2.1. Study Area

The Verde River Basin (VRB) is situated in the southern part of the state of Minas Gerais (Figure 1), making it a crucial region with regard to biodiversity conservation. With a drainage area of approximately 4100 km$^2$ and an altitude range of 809 m to 2742 m, the VRB encompasses 31 municipalities, of which 16 are entirely contained within the basin. Furthermore, the VRB has a predominantly Cwb climate, and its average annual temperature is 18 °C [22,23].

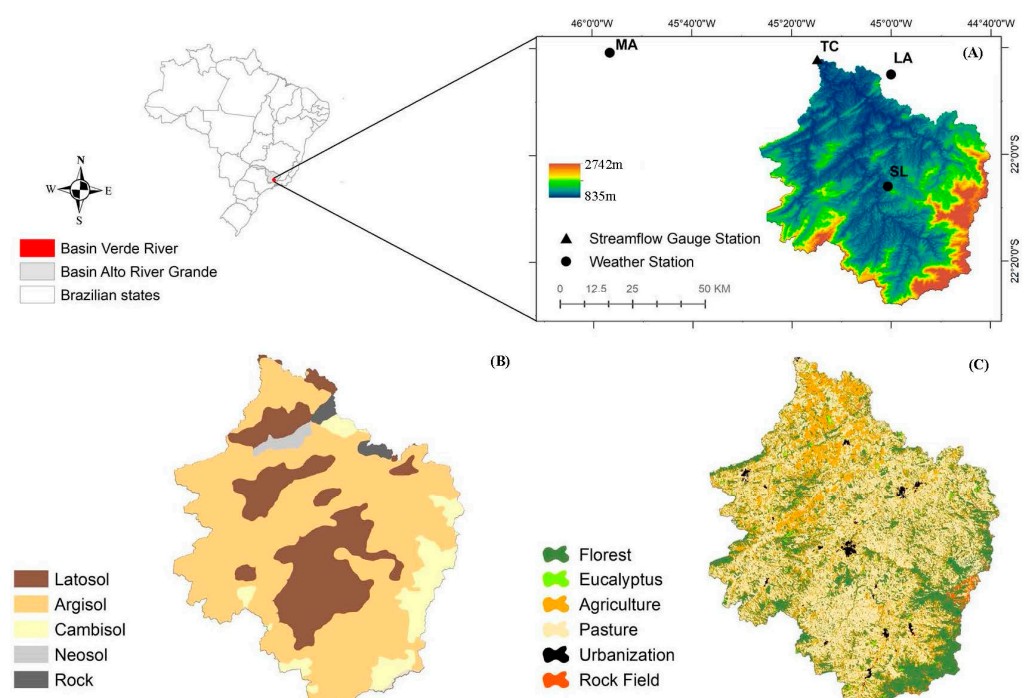

**Figure 1.** Verde River Basin in the state of Minas Gerais, Brazil; digital terrain model. (**A**) Climate stations' locations; (**B**) soil map; (**C**) land use and land cover map.

The land use classification considered the MapBiomas Collection 5.0 data [24], which include annual land cover data in Brazil from 1985 to 2019. Land cover in the VRB consists of pasture (69.2%), native forest (21.3%), rock fields (1.7%), eucalyptus plantations (0.2%), agriculture (7%), and urbanized landscapes (0.6%), as shown in Figure 1C. Although a significant portion of the land use in the VRB (Verde River Basin) is dedicated to pastures, the region's economy revolves around coffee production; dairy farming; poultry farming; temporary crops such as corn, beans, and rice; tourism; and hydroelectric energy production.

Regarding the soil classes in the VRB, Argisol (65.3%), Latosol (23.3%), and Cambisol (8.9%) predominate (Figure 1B), with rocky (1.3%) and Neosol (1.2%) being less common. The basin's predominant land-use and land-cover patterns are as follows: pasture (69.2%), native forest (21.3%), rocky field (1.7%), eucalyptus plantation (0.2%), agriculture (7%), and urbanized land (6%).

For each specified soil class in the watershed, input information such as soil depth ($Z$), soil moisture at saturation point ($\theta_s$), and soil moisture at permanent wilting point ($\theta_{pwp}$) was incorporated. Table 1 presents the values of $Z$, $\theta_s$, and $\theta_{pwp}$ according to the literature.

**Table 1.** Values adopted for depth, saturation point moisture ($\theta_s$), and permanent wilting point moisture ($\theta_{pmp}$) for each soil class in the Rio Verde watershed.

| Soil Class | Depth (cm) | $\theta_s$ (m$^3$m$^{-3}$) | $\theta_{pwp}$ (m$^3$m$^{-3}$) |
|---|---|---|---|
| Cambisol (CX) | 100 | 0.597 | 0.171 |
| Latosol (LVA) | 110 | 0.555 | 0.240 |
| Argisol (PVAD) | 80 | 0.45 | 0.19 |
| Neosol (RY) | 20 | 0.574 | 0.183 |
| Rock (AR) | 0 | 0 | 0 |

Source: Adapted from Junqueira Junior [25].

The vegetation attributes required by the models include leaf area index (LAI), rooting depth, albedo, surface resistance, and height. Table 2 presents the parameters extracted

from the literature for each vegetation cover class. By utilizing land use map information, the system allows for the identification of parameters associated with vegetation required by the LASH hydrological model, namely, albedo, vegetation canopy height, surface resistance, rooting depth, and leaf area index.

**Table 2.** Parameters extracted from the literature for each vegetation cover class and their respective references.

| Vegetation | LAI $m^2m^{-2}$ | Height (m) | Albedo | Surface Resistance $(sm^{-1})$ | Rooting Depth (mm) |
|---|---|---|---|---|---|
| Agriculture | 0.3–7.0 | 0–1.52 | 0.15–0.20 | 40 | 500 |
| Pasture | 1.86–3.99 | 0.5 | 0.20–0.26 | 70 | 600 |
| Florest | 6.25 | 10 | 0.13–0.18 | 100 | 2000 |
| Cerrado | 1.9 | 5 | 0.13–0.18 | 150 | 2000 |
| Eucalyptus | 3.5 | 5 | 0.13–0.18 | 100 | 1500 |
| RockField | 0 | 0 | 0.10–0.35 | 545.3 | 500 |

Source: Viola [26].

### 2.2. Hydrology Models

The Lavras Simulation of Hydrology (LASH) model is a semi-distributed hydrological model that uses the modified SCS-CN model [27] as its basis. The model incorporates the equation of Brooks and Corey to estimate subsurface flows and base flows along with the Muskingum-Cunge linear model for drainage network routing [28]. LASH operates on the principle of the water balance equation (Equation (1)):

$$\frac{A_t - A_{t-1}}{\Delta t} = \frac{[(P + D_t - ET_R - D_B - D_{SS} - D_S)]}{\Delta t} \tag{1}$$

where $A_t$ denotes current soil water storage at time $t$; $A_{t-1}$ denotes soil water storage in the time interval immediately preceding $t$; $P$ is average precipitation (discounted from interception (*IT*)); $D_t$ denotes capillary rise; $ET_R$ is the actual evapotranspiration; $D_B$ denotes base runoff; $D_{SS}$ denotes subsurface runoff; $D_S$ denotes surface runoff; and $\Delta t$ is the time interval.

Variable Infiltration Capacity (VIC) is a distributed hydrological model comprising two modules: (i) a rainfall runoff transformation module [10–12] and (ii) a base discharge routing module [29,30] that utilizes linearized Saint–Venant equations to propagate base discharge. The VIC model's underlying principle is based on the variable infiltration curve (Equation (2)):

$$i = i_m * \left[ 1 - (1 - A)^{1/b_i} \right] \tag{2}$$

where $i$ is the point infiltration capacity and $i_m$ is the maximum soil infiltration capacity; $A$ is the fractional soil area with an infiltration capacity less than $i$; and $b_i$ is the infiltration shape parameter.

The Distributed Hydrological Model of the National Institute for Space Research (MHD-INPE) was developed by Rodriguez and Tomasella [16] and Siqueira Júnior et al. [31], and it comprises four modules: (i) vertical water balance in the soil; (ii) evapotranspiration; (iii) surface, subsurface, and groundwater flows; and (iv) propagation in channels. The MHD-INPE model considers the following variables: groundwater flux ($Q_{sub}$) estimated via Equation (3), subsurface flux ($Q_{ss}$) from this layer estimated via Equation (4), and aquifer recharge ($Q_r$) estimated via Equation (5).

$$Q_{Sub} = \frac{T_{sub} * tan\beta}{\lambda_\mu^\mu} \left[ 1 - \frac{S_{max} - S_t}{S_{max}(1 - \xi)} \right]^\mu = \frac{T_{sub} \, tan\beta}{\lambda_\mu^\mu} \left[ \frac{S_t - \xi S_{max}}{S_{max}(1 - \xi)} \right]^\mu \tag{3}$$

$$Q_{ss} = \frac{\alpha\, D_1 K_{ss} \tan\beta}{\lambda_\eta^\eta} \left(\frac{SS_t}{SS_{max}}\right)^\eta \tag{4}$$

$$Q_r = K_{ss}\left(\frac{SR_t}{SRmax}\right)^\eta \tag{5}$$

where $T_{sub}$ is the transmissivity when the water table is at the surface; $\tan\beta$ is the slope of the local topography; $\mu$ is a parameter that defines the shape of the relationship between transmissivity and soil depth; $\xi$ is the drainable porosity; $S_{max}$ is the maximum capacity of water storage in the cell; and $\lambda\mu$ is the average contribution area per contour unit, assuming a potential profile for transmissivity. The topographic parameter, $\lambda\mu$, is solved independently for each regular grid cell. $S_{max}$ is solved for the bottom soil layer in the model while considering its depth, $D3$, and soil porosity, $\varphi$. $K_{SS}$ is the saturated hydraulic conductivity in the upper soil layer; $D_1$ is the thickness of the top soil layer; $SS_{max}$ is the maximum water storage capacity in this layer; $\eta$ is the Brooks–Corey parameter; $\alpha$ accounts for the soil anisotropy in the equation; $SS_t$ is the average water storage in the upper layer; $\lambda\eta$ is the average value of the contribution area per contour unit raised to the $1/\eta$ potency; $SR_{max}$ is the maximum storage capacity in the middle tier; and $SR_t$ is the average storage in the middle tier grid at time $t$.

*2.3. Observed Data*

Meteorological data were obtained from the National Institute of Meteorology [32] and were taken from three conventional stations (Figure 1A): São Lourenço station (SL; $22°\,07'48''$ S and $45°\,02'24''$ W) located within the VRB and Machado (MA; $21°\,40'48''$ S; $45°\,56'24''$ W) and Lavras (LV; $21°\,13'34''$ S, $44°\,58'47''$ W) stations located outside the basin's boundaries. Fluviometric data were collected from the Três Corações station (TC; $-21°\,42'11''$ S and $45°\,14'51''$ W) by the National Water and Basic Sanitation Agency. The LASH, VIC, and MHD models require the maximum and minimum daily temperature (°C), wind speed (ms$^{-1}$), precipitation (mm), global solar radiation (MJ/day.m$^{-2}$), relative humidity, dew point temperature (°C), and surface-level atmospheric pressure (*mb*) as inputs. Therefore, we used meteorological and discharge gauge station data for the period from 1990 to 2005 for prospection.

A Digital Elevation Model was obtained from the Shuttle Radar Topography Mission (SRTM), providing altitudes ranging from 835 to 2742 m for the VRB. The spatialization of the basin's soil classes was determined based on the new map of Brazil [33]. To input the required vegetation parameters in the LASH, VIC, and MHD models, we gathered data from the literature [19–21,28,34–40] on leaf area index, root depth, albedo, surface resistance, and height. Notably, the vegetation parameters were kept constant throughout the model calibration process.

In terms of the LASH model, we divided the study basin into 57 sub-basins, each with a drainage area ranging from 1.14 km$^2$ to 215.65 km$^2$. We defined upper and lower limits for each parameter during calibration using the SCE-UA algorithm [41], as outlined by Beskow et al. [15]. The model's seven most-sensitive parameters were identified as follows: (*a*) the initial abstraction coefficient ($\lambda$); (*b*) hydraulic conductivity of the shallow saturation zone reservoir (*KB*); (*c*) hydraulic conductivity of the subsurface reservoir (*KSS*); (*d*) maximum flow returning to soil via capillary rise (*KCR*); (*e*) response time parameter of the surface reservoir (*CS*); (*f*) response time parameter of the subsurface reservoir (*CSS*); and (*g*) baseflow recession time (*CB*). The $\lambda$ parameter is directly linked to initial rainfall abstraction, which corresponds to the portion of rainfall loss that occurs before direct surface runoff generation [15]. We emphasize that the results of this prospection are published in this scientific article for further analysis and discussion.

For VIC modeling, the basin's surface was represented by grid cells, for which a resolution of 0.01° was defined (3768 grid cells). VIC calibration was conducted manually by changing each parameter individually according to the methods of Gao et al. [42] and

Liang et al. [10]. The following parameters were considered for calibration: *Ds* (fraction of maximum baseflow velocity where non-linear baseflow begins), *Ws* (fraction of maximum soil moisture where non-linear baseflow occurs), *bi* (variable infiltration curve), *h3* (thickness of the third layer), *C* (kinematic wave celerity), and *D* (kinematic wave diffusion coefficient).

For MHD, a resolution of 0.0450 was defined (166 grid cells). The automatic calibration process utilized the Hydrologic Response Unit (HRU) derived from the combination of land-use and soil type data [17]. Calibration was performed using the Strength Pareto Evolutionary Algorithm-SPEA2 method [43]. Ten parameters were adjusted during the calibration process, including soil layer depths (D1, D2, and D3), the hydraulic conductivity multiplier of the upper layer, maximum transmissivity of the bottom layer ($T_{sub}$), the decay of transmissivity with the thickness of the saturated zone ($\mu$), the ratio of field capacity to porosity ($\xi$), the coefficient of anisotropy ($\alpha$), and the routing water storage parameter for surface and subsurface flows ($C_{sup}$) and baseflow ($C_{sub}$).

### 2.4. Application, Calibration, and Validation of the Hydrological Models

A hydrological simulation in the VRB using LASH, VIC, and MHD was conducted for the period between 1990 and 2005, for which calibration was performed between 1993 and 1999 and validation was performed between 2000 and 2005. The initial hydrological conditions had uncertainties; therefore, the first three years were considered to be a warm-up period. The simulated discharges were compared with the observed discharges using statistical indices such as the Nash–Sutcliffe index [44], the logarithmic version of LNASH, $R^2$ [45], and *PBIAS* [46] (Equations (6)–(9), respectively):

$$NASH = 1 - \frac{\sum_{i=1}^{n}\left(Q_{oi} - Q_{si}\right)^2}{\sum_{i=1}^{n}\left(Q_{oi} - Q_O\right)^2} \tag{6}$$

$$LNASH = 1 - \frac{\sum_{i=1}^{n}\left(\log(Q_{OI}) - \log(Q_{si})\right)^2}{\sum_{i=1}^{n}\left(\log(Q_{OI}) - \log(Q_O)\right)^2} \tag{7}$$

$$R^2 = \left\{ \frac{\sum_{i=1}^{n}\left(Q_{oi} - \overline{Q}_0\right)\left(Q_{si} - \overline{Q_s}\right)}{\sum_{i=1}^{n}\left[\left(Q_{oi} - \overline{Q_o}\right)^2\right]^{0,5}\left[\left(Q_{si} - \overline{Q_s}\right)^2\right]^{0,5}} \right\}^2 \tag{8}$$

$$PBIAS = \frac{\sum_{i=1}^{n} Qsi - \sum_{i-1}^{n} Qoi}{\sum_{i=1}^{n} Qoi} * 100 \tag{9}$$

where $Q_{si}$ is the simulated discharge at time $I$, $Q_{oi}$ is the observed discharge at time i, $\overline{Q}_o$ is the average observed discharge, $\overline{Q}_s$ is the average simulated discharge, and $n$ denotes the total number of observed data.

The NASH coefficient is used to determine a model's ability to reproduce observed discharge series and has a strong influence on maximum discharges, whereas LNASH is more influenced by minimum discharges. The $R^2$ value indicates the degree of correlation between simulated and observed values [16,17,47]. *PBIAS* measures the average tendency of the simulated discharge to either overestimate or underestimate the observed discharge [45]. *PBIAS* values close to zero indicate that a model shows no tendencies, negative values indicate underestimated discharges, and positive values indicate overestimated discharges [46]. Generally, hydrological model performance can be classified according to the information in Table 3 proposed by Moriasi et al. [48].

**Table 3.** Classification of statistical indices for evaluating hydrological models' performance.

| Statistical Indices | Range | Performance Classification | | | |
|---|---|---|---|---|---|
| | | Very Good | Good | Satisfactory | Unsatisfactory |
| NASH and LNASH | $-\infty$–1 | >0.80 | 0.70–0.80 | 0.50–0.70 | $\leq$0.50 |
| $R^2$ | 0–1 | >0.85 | 0.75–0.85 | 0.60–0.75 | <0.60 |
| PBIAS | $-\infty$–100 | <$\pm$5% | $\pm$5–$\pm$10% | $\pm$10–$\pm$15% | >$\pm$15% |

Source: adapted from Moriasi et al. [48].

## 3. Results and Discussion

Table 4 presents the optimized parameters utilized in the VIC, LASH, and MHD hydrological models. The parameters used for calibration were determined to be the most sensitive based on the studies by Gao et al. [42] and Liang et al. [10] for VIC, Beskow et al. [15] for LASH, and Rodriguez and Tomasella [16] and Siqueira Júnior et al. [31] for MHD. Following calibration, a validation process was carried out by utilizing the optimized parameters obtained in the calibration phase.

**Table 4.** Calibration parameters of the VIC, LASH, and MHD hydrological models as well as their physical meaning and final values.

| Parameter | Description | Unit | Range | Final Values |
|---|---|---|---|---|
| | VIC | | | |
| $b_i$ | Variable Infiltration Curve | - | 0.001–0.4 | 0.35 |
| $D_S$ | Fraction of maximum velocity of baseflow where non-linear baseflow begins | Fraction | 0.001–0.99 | 0.01 |
| $W_S$ | Fraction of maximum soil moisture where non-linear baseflow occurs | Fraction | 0.001–0.99 | 0.05 |
| $H_3$ | Thickness of the third layer | m | 0.05–2 | 0.5 |
| $C$ | Kinematic wave celerity | m s$^{-1}$ | 0.5–3 | 0.5 |
| $D$ | Kinematic wave diffusion coefficient | S m$^{-1}$ | 200–400 | 2200 |
| | LASH | | | |
| $\lambda$ | Initial abstraction coefficient | - | 0.01–0.5 | 0.07 |
| $K_B$ | Hydraulic conductivity of shallow saturates zone reservoir | mm day$^{-1}$ | 0.0–6 | 3.33 |
| $K_{SS}$ | Hydraulic conductivity of subsurface reservoir | mm day$^{-1}$ | 0–250 | 245.41 |
| $K_{CR}$ | Maximum flow returning to soil via capillary rise | mm day$^{-1}$ | 0–5 | 1.03 |
| $C_S$ | Response time parameter of the surface reservoir | - | - | 84.45 |
| $C_{SS}$ | Response time parameter of the sub-surface reservoir | - | - | 16,677.22 |
| $C_B$ | Baseflow recession time | day | - | 105.46 |
| | MDH-INPE | | | |
| $D_1$ | Thickness of the upper layer | m | 0–10 | 3.8 |
| $D_2$ | Thickness of the intermediate layer | m | 0–10 | 0.35 |
| $D_3$ | Thickness of the bottom layer | m | 0–30 | 0.09 |
| $K_{SS}$ | Saturated of hydraulic conductivity | mm day$^{-1}$ | 0.01–10 | 0.37 |
| $T_{sub}$ | Maximum transmissivity of the bottom layer | m$^2$ day$^{-1}$ | 0.01–1000 | 0.48 |
| $\mu$ | Decay of transmissivity with the thickness of the saturated zone | - | 0.01–4 | 0.0005 |
| $C_{sup}$ | Routing water storage parameter for surface and subsurface flows | day | 0.001–10 | 25.39 |
| $C_{sub}$ | Routing water storage parameter for baseflow | day | 0.001–2000 | 272.4252 |
| $\zeta$ | Ratio of field capacity to porosity | - | | |
| $\alpha$ | Coefficient of anisotropy | - | 1–10,000 | 394.7182 |

Table 5 presents the accuracy statistics results used to evaluate the performance of the three hydrological models (LASH, VIC, and MHD-INPE). The results indicate that

the LASH and MHD-INPE models performed better than the VIC model. All statistical indices demonstrated satisfactory performance in terms of both calibration and validation according to the classification proposed by Moriasi et al. [48]. The performance analysis of the VIC model revealed that there was a significant challenge in simulating discharges during the dry period, yielding LNASH values below 0.50. Although the PBIAS values were satisfactory, they indicated a tendency to underestimate the observed discharge in both calibration and validation. Overall, the NASH values indicated good performance for all three hydrological models in simulating peak discharges, with the LASH and MHD-INPE models demonstrating the best simulated peak discharge values.

**Table 5.** Statistical indices of the hydrological models regarding daily discharge.

| Statistical Indices | Calibration | | | Validation | | |
|---|---|---|---|---|---|---|
| | **LASH** | **VIC** | **MHD** | **LASH** | **VIC** | **MHD** |
| NASH ($C_{NS}$) | 0.85 | 0.79 | 0.79 | 0.80 | 0.77 | 0.87 |
| LNASH ($logC_{NS}$) | 0.89 | 0.22 | 0.84 | 0.81 | 0.35 | 0.86 |
| $R^2$ | 0.85 | 0.85 | 0.79 | 0.85 | 0.85 | 0.88 |
| $P_{BIAS}$ | 0.60 | −14.85 | 0.80 | 6.80 | −14.48 | −7.30 |

Table 5 shows that the hydrological models MHD and LASH achieved satisfactory results in relation to the VRB. Similar results were also reported by other authors who evaluated hydrological model performance in the southern region of the state of Minas Gerais. Melo et al. [19] obtained satisfactory results at a daily time step using the MHD-INPE and DHSVM models, reporting NASH, LNASH, and *Pbias* values of 0.80, 0.75, and −8.0, respectively. Zackia et al. [20] also reported NASH values above 0.7 for MHD-INPE for calibration and validation at a daily time step. Viola et al. [49] applied LASH to four headwater catchments (Aiuruoca River, Grande River, Sapucai River, and Verde River) and reported NASH values between 0.7 and 0.86 for monthly simulations.

Figures 2 and 3 present the daily discharge simulated by LASH, VIC, and MHD, as well as the observed discharge, for both the calibration and validation periods. The recession periods demonstrated that the hydrological models MHD and LASH successfully reproduced the storage processes present in the watershed, as indicated by the recession curve extending over time intervals equivalent to the observed flows. This resulted in a very good evaluation of the *logCNS* coefficient. The results of this coefficient confirmed the models' high reliability in simulating the recession periods of the hydrograph according to the classifications by Moriasi et al. [48]. In contrast, the simulated flows produced using VIC could not accurately represent the observed data during the dry period, indicating a certain limitation and underestimation of flows during the recession period. The adjustment of the VIC hydrological model may be related to the inherent difficulties of the manual calibration process, which requires longer processing times.

In Figure 3, during the validation phase, the hydrological models MHD-INPE, LASH, and VIC showed good agreement between simulated and observed data as assessed via the NASH coefficient (*CNS*), which is associated with the efficiency of peak flow estimation in the hydrograph. The following results were obtained: 0.87, 0.80, and 0.77, respectively, which are considered very good according to the classification scheme devised by Moriasi et al. [48] When analyzing the minimum flows, *logCNS* only yielded satisfactory results for the MHD-INPE and LASH models, which were able to reproduce the observed daily flows. Therefore, the results obtained for the VRW confirm that the MHD-INPE and LASH models were able to adequately represent the values of minimum and maximum flows during the calibration and validation periods. The obtained *Pbias* values were −7.30%, 6.80%, and 14.48% for MHD-INPE, LASH, and VIC, respectively, indicating satisfactory performance. Similar results were obtained by Carvalho et al. [5], who achieved good performance using the LASH model, and Melo et al. [19], who used the MHD-INPE model in a headwater watershed.

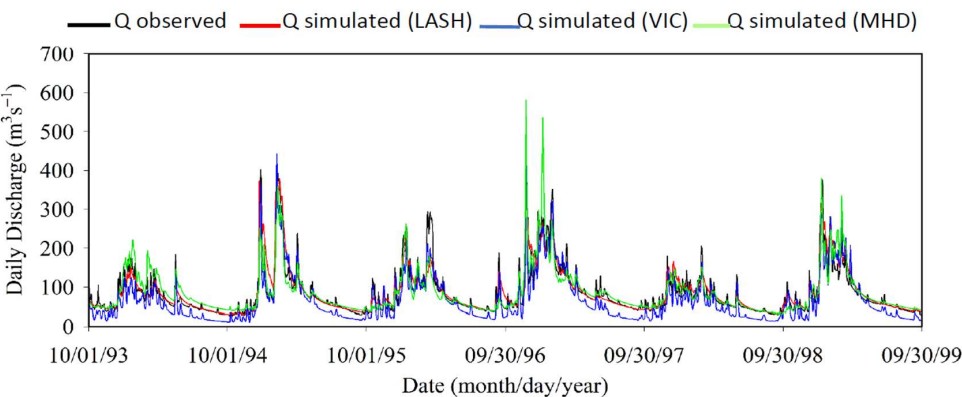

**Figure 2.** Daily discharge observed and simulated via LASH, MHD, and VIC during the calibration process.

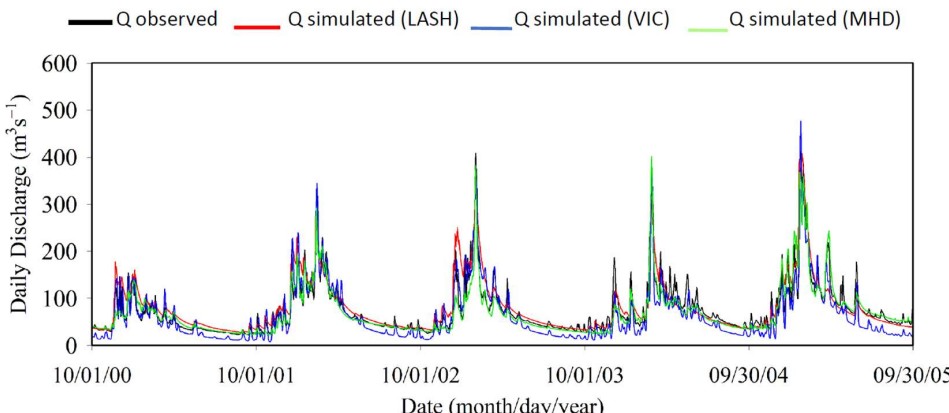

**Figure 3.** Daily discharge observed and simulated via LASH, MHD, and VIC during the validation.

Overall, when compared to the observed data, the daily discharge simulated using the LASH and MHD-INPE models showed better results than those simulated via VIC. The poorer performance of VIC may be attributed to the manual calibration process used, as this calibration method requires a significant amount of processing time to achieve the best parameter combination, which is assessed individually. Moreover, manual calibration requires an inter-parameter analysis, which is typically performed by fixing one parameter and altering the others to determine which parameter is more sensitive. In general, when comparing the adjustment of the observed and simulated values, the result was better for the MHD-INPE and LASH models compared to that of the VIC model. These differences in the simulated flow results obtained using the MHD-INPE, LASH, and VIC models indicate the need to assess the uncertainties of hydrological simulations obtained from different hydrological models. Singh and Marcy [50] and Orth et al. [6] emphasized that performance evaluation across different hydrological models can reduce uncertainties stemming from the different structures and complexities of the chosen hydrological models.

Figure 4 depicts the behavior of the flow duration curve for the observed and simulated flows provided by the three models used. The flow duration curve provides an estimate of the time frequency at which a given flow is equaled or exceeded. In this context, considering water resources management, it is an important tool for representing the frequency of maximum and minimum reference flows simulated by the hydrological models. It can be observed that both models exhibited a frequency distribution similar to that of the observed data. The MHD-INPE and LASH models demonstrated better performance and closely approximated the observed flow values for lower flows. On the other hand, it is evident that the minimum flows from the curve representing the VIC model deviated from the observed flows. Overall, the VIC model tended to underestimate the flow values.

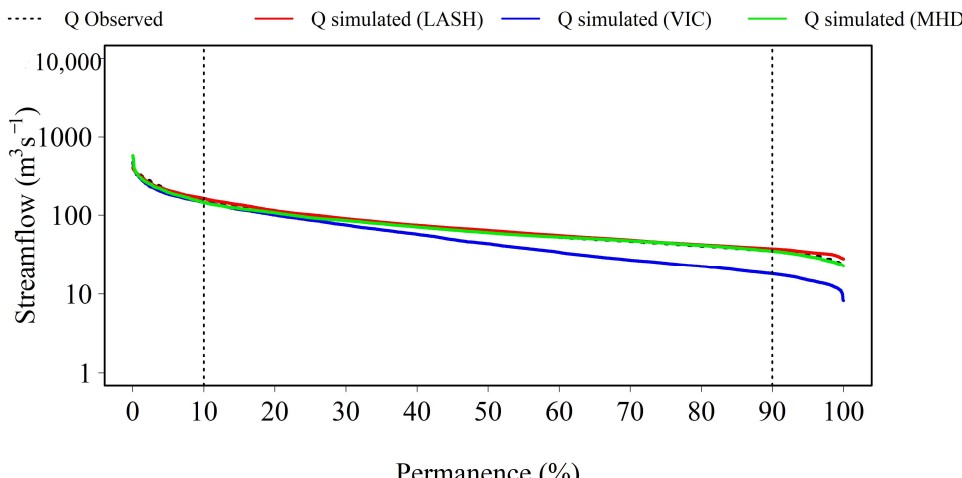

**Figure 4.** Permanence curve of the observed and simulated discharges (LASH, MHD, and VIC).

The minimum discharges estimated by the VIC model were found to be underestimated compared to the observed discharges. For instance, the observed $Q_{90}$ (the flow that was exceeded or matched 90% of the time) value was 35.03 $m^3s^{-1}$, while the simulated $Q_{90}$ discharges for the LASH, VIC, and MHD models were 43.87 $m^3s^{-1}$, 18.03 $m^3s^{-1}$, and 35.89 $m^3s^{-1}$, respectively. Similarly, the observed $Q_{10}$ (the flow that was exceeded or matched 10% of the time) value was 151.61 $m^3s^{-1}$, while the $Q_{10}$ simulated using LASH, VIC, and MHD were 162.78 $m^3s^{-1}$, 144.84 $m^3s^{-1}$, and 147.17 $m^3s^{-1}$, respectively. Overall, the permanence curve of the daily discharges simulated by LASH and MHD provided the best fit. The MHD-INPE and LASH models have been assessed in several studies regarding their accuracy in estimating the flow duration curve, consistently demonstrating good performance with minimal discrepancies in minimum flows [15,19,35].

Figure 5 displays the cumulative daily runoff data, with the data simulated by LASH and MHD showing high levels of agreement when compared to the observed data. The cumulative runoff values were 7.036 mm (observed), 7.252 mm (simulated via LASH), 7.016 mm (simulated via MHD), and 5.892 mm (simulated via VIC). At the end of the evaluation period, the values simulated via LASH were slightly above the observed values. On the other hand, the values of the data simulated using VIC were generally below those of the observed data.

Table 6 displays the results of the observed and simulated annual vertical water balance components obtained using the LASH, VIC, and MHD hydrological models. The simulated evapotranspiration-to-precipitation ratios averaged from 44 to 57%, 53 to 78%, and 37 to 64% for LASH, VIC, and MHD, respectively. The higher average values of annual evapotranspiration simulated via VIC may indicate an underestimation of runoff. These findings align with the study by Alvarenga et al. [7], which showed reasonable performance in discharge simulation in the Verde River basin using SWAT and VIC. The authors reported average annual values of evapotranspiration ranging from 57.9 to 65.7% of precipitation. The cited paper revealed that VIC outperformed SWAT. In contrast, this study demonstrated that LASH and MHD performed better than VIC.

**Table 6.** Annual water balance components (1993–2005) in the VRB.

| YEAR | P (mm) | LASH | | | | VIC | | | | MHD | | | |
|---|---|---|---|---|---|---|---|---|---|---|---|---|---|
| | | ET (mm) | P-ET (mm) | ET/P (mm) | | ET (mm) | P-ET (mm) | ET/P (mm) | | ET (mm) | P-ET (mm) | ET/P (mm) | |
| 1993 | 1451.60 | 733.93 | 717.67 | 0.51 | | 841.37 | 610.23 | 0.58 | | 764.67 | 686.93 | 0.53 | |
| 1994 | 1348.97 | 653.72 | 695.26 | 0.48 | | 888.92 | 460.05 | 0.66 | | 754.17 | 594.80 | 0.56 | |
| 1995 | 1504.59 | 744.24 | 760.35 | 0.49 | | 995.91 | 508.67 | 0.66 | | 772.98 | 731.61 | 0.51 | |

**Table 6.** *Cont.*

| YEAR | P (mm) | LASH | | | VIC | | | MHD | | |
|---|---|---|---|---|---|---|---|---|---|---|
| | | ET (mm) | P-ET (mm) | ET/P (mm) | ET (mm) | P-ET (mm) | ET/P (mm) | ET (mm) | P-ET (mm) | ET/P (mm) |
| 1993 | 1451.60 | 733.93 | 717.67 | 0.51 | 841.37 | 610.23 | 0.58 | 764.67 | 686.93 | 0.53 |
| 1994 | 1348.97 | 653.72 | 695.26 | 0.48 | 888.92 | 460.05 | 0.66 | 754.17 | 594.80 | 0.56 |
| 1995 | 1504.59 | 744.24 | 760.35 | 0.49 | 995.91 | 508.67 | 0.66 | 772.98 | 731.61 | 0.51 |

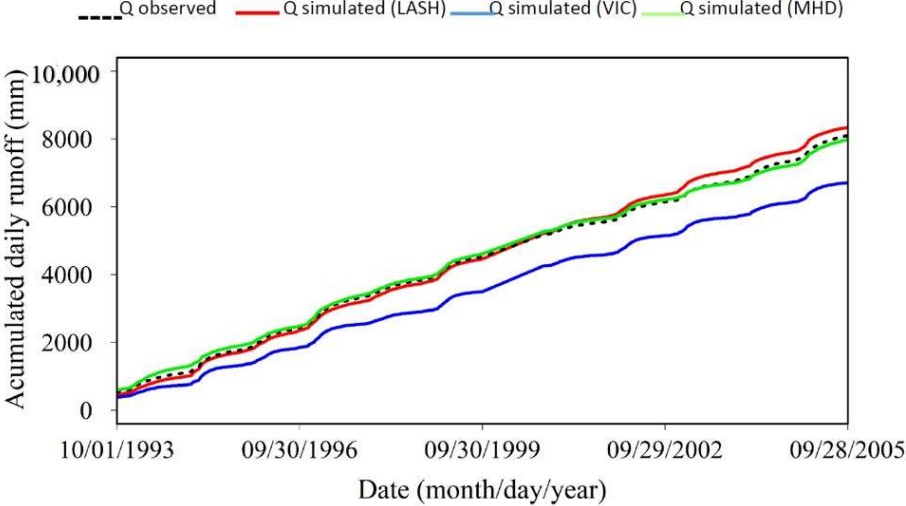

**Figure 5.** Cumulative runoff in the VRB.

## 4. Conclusions

A performance analysis of the LASH, MHD, and VIC hydrological models was conducted in the Verde River basin. The models' performance can vary greatly depending on the hydrological conditions, such as floods and droughts. Overall, the hydrological models were able to simulate the average daily discharges in the basin. During calibration and validation, LASH and MHD demonstrated better performance when compared to VIC, which encountered significant problems in simulating the dry period. As a result, the results of LNASH were less than 0.5 in this case.

Several aspects regarding the results obtained via VIC reflect the calibration type (manual and with defined limits), as well as uncertainties related to different model structures, which led to unsatisfactory results. Nevertheless, the model can potentially be used as a tool for the prediction of peak flows and may be important for decisions regarding support for flood management.

The findings reveal that LASH and MHD simulations effectively replicated discharges with a remarkable degree of accuracy, rendering them invaluable tools for water resource management in the studied basins. Moreover, these outcomes can serve as a foundation for forthcoming research focused on food and water security, particularly in relation to climate change scenarios. Consequently, these models hold significant promise for advancing our understanding and preparedness with respect to addressing potential challenges in the study region and beyond.

**Author Contributions:** Conceptualization, C.d.M.M.d.O. and L.A.A.; Methodology, C.d.M.M.d.O., L.A.A., S.B., Z.A.d.C., M.M.V., P.A.M., J.T., A.C.N.S. and V.S.O.C.; Validation, C.d.M.M.d.O., L.A.A., S.B., Z.A.d.C., M.M.V. and J.T.; Formal Analysis, C.d.M.M.d.O., L.A.A., V.S.O.C. and V.O.S.; Investigation, C.d.M.M.d.O.; Writing—Original Draft Preparation, C.d.M.M.d.O. and L.A.A.; Writing—Review and Editing, C.d.M.M.d.O., L.A.A., S.B., Z.A.d.C., M.M.V., P.A.M., J.T., A.C.N.S., V.S.O.C. and V.O.S.; Supervision, L.A.A. All authors have read and agreed to the published version of the manuscript.

**Funding:** This research received no external funding.

**Data Availability Statement:** The data are not publicly available.

**Conflicts of Interest:** The authors declare no conflict of interest.

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
