# Peer review of "Hydrological Model Performance in the Verde River Basin, Minas Gerais, Brazil"

_resources, doi:10.3390/resources12080087_

Round 1

Reviewer 1 Report

The manuscript titled " HYDROLOGICAL MODEL PERFORMANCES IN THE VERDE RIVER BASIN, MINAS GERAIS, BRAZIL" was reviewed. The aim of this manuscript is to investigate uncertainties between hydrological models. The structure of the manuscript is well organized. However, the discussion part is very weak, and the significance of the results needs to be further proved. 

Why have the authors selected LASH, VIC, and MHD, hydrological models? Are there any selection criteria? Why not included SWAT and HEC-HMS, etc.

The Verde River basin has a drainage area of 4100 km², and only data from two weather stations were used in the study. The authors should consider including data from other nearby stations as well.

In Figure 2, all the models overestimated the low discharge. Why did this occur?

Please add a scatter plot between simulated and observed discharge.

The discussion section of the manuscript is weak. Please address this issue

 Minor editing of English language required

Author Response

REVIEWER #1:

Why have the authors selected LASH, VIC, and MHD, hydrological models? Are there any selection criteria? Why not included SWAT and HEC-HMS, etc.

This research is part of a larger project called “Hydrological Simulation and Evaluation of the Impacts of Climate Change and Land Use in the Verde River Watershed in the Minas Gerais State”. with several partner institutions such as the Federal University of Lavras (UFLA), the National Center for Monitoring and Early Warning of Natural Disasters (CEMADEN), and the State University of Campinas (UNICAMP). The mention institutions have been developing and applying the hydrological models LASH, MHD, and VIC over the years (Adicionar publicações). As a result, the goal of this work was to evaluate the application of these models over the Verde River basin under climate change and land use scenarios.

The Verde River basin has a drainage area of 4100 km², and only data from two weather stations were used in the study. The authors should consider including data from other nearby stations as well.

Meterological data with high quality is very difficult to be found in Brazil. For this specific work, the authors have chosen all the data available over the Verde River basin and around it. The Passa Quatro weather station haven’t filled the criteria to be in this work. The following picture show the metereological statitions in the Minas Gerais state where the River Basin is included.

In Figure 2, all the models overestimated the low discharge. Why did this occur?

In this calibration phase, specifically during the recession period, the hydrological models MHD and LASH closely followed the low discharges during the dry season, while only VIC exhibited a different behavior, underestimating the discharge.

 In the following picture, is possible to verify that the models don’t overestimate the low discharges.

Please add a scatter plot between simulated and observed discharge

The authors gently disagree with this suggestion given by the reviewer. The main point of the scatter plot is to evaluate the quality of the simulation, comparing the observed and the simulated data. In this present work, the author have chosen to demonstrate the quality of the simulations using the accumlated data. The results can be seen in Figure 5.

The discussion section of the manuscript is weak. Please address this issue

                The discussion section was changed.

Reviewer 2 Report

The paper “Hydrological Model Performances in the Verde River Basin, Minas Gerais, Brazil” presents simulations and analysis performed using different hydrological models to simulate the discharge values in the Verde River Basin.

The paper presents the models and then the metrics used to estimate the goodness of fit and the parameters values used to calibrate the models. Despite the interesting topic, the models are already available from literature and thus it is not clear what is the original contribution of this paper to the scientific community.

The Authors should clearly present the original contribution. Moreover, the paper would substantially benefit from a deeper explanation of the parameters, their range of variation should be explained (e.g. type of soil, characteristic values) so that the reader could better understand results and Authors’ choice.

The models are characterised by many parameters, a sensitivity analysis would be needed to test which are the most relevant for the model and to characterise the behaviour of the basin. Also, to many parameters correspond a large uncertainty, so reducing the parameters number would decrease the uncertainty as well.

Could the Authors please specify if they are using the peak discharge values, or the time series of a specific flood event and the length of the data used for estimating the metrics in equations 6-9?

I kindly disagree with the sentence “Nevertheless, the model (VIC) has the potential for use as a tool in modelling.” P.11 as the VIC model showed an overall not good performance. Would it be better to use it to simulate peak discharge values? Could please the Authors better explain their idea?

The time series used to calibrate the model is short, however the performance is promising. I’d kindly suggest increasing the time series length for a more robust validation.

In substance, I am sorry to say that the paper is not ready to be published and unless the original contribution is presented and the analysis are largely improved, I cannot see the possibility of publishing it.

I suggest a major revision to provide the Authors the chance to highly improve the manuscript.

The English language is good and minor editing may be fixed

Author Response

Manuscript ID: resources-2428444

Hydrological Model Performances in the Verde River Basin, Minas Gerais, Brazil

Conceição de M. M. de Oliveiraa,b ; Lívia A. Alvarengaa ; Samuel Beskowb ; Zandra Almeida da Cunhab; Marcelle Martins Vargasb ; Pâmela A. Meloa ; Javier Tomasellad ; Ana Carolina N. Santosd ; Vinicius S. O. Carvalhoe, Vinicius O. Silvaa

Answer to the Editorss

In order to adequate our work to the suggestions of the reviewers, we follow all the comments. The authors have appreciated the reviewer's valuable suggestions The answers can be observed below:

REVIEWER #2:

The paper presents the models and then the metrics used to estimate the goodness of fit and the parameters values used to calibrate the models. Despite the interesting topic, the models are already available from literature and thus it is not clear what is the original contribution of this paper to the scientific community.

was added to the text

The Authors should clearly present the original contribution. Moreover, the paper would substantially benefit from a deeper explanation of the parameters, their range of variation should be explained (e.g. type of soil, characteristic values) so that the reader could better understand results and Authors’ choice.

was added to the text

The models are characterised by many parameters, a sensitivity analysis would be needed to test which are the most relevant for the model and to characterise the behaviour of the basin. Also, to many parameters correspond a large uncertainty, so reducing the parameters number would decrease the uncertainty as well.

The simulation and calibration period are diffferent because there was intense land uses changes during the whole period and because a intense drought period affected the basin at the beginning of 2000 (Cavalcanti e Kousky 2001). Therefore, the study period is considered a rigorous test for all models in terms of capturing hydrological variability. Because of this we did not include sensitivity analysis. Nevertheless, in the conclusions section a specific comment was included together with the need of an extended series.

Could the Authors please specify if they are using the peak discharge values, or the time series of a specific flood event and the length of the data used for estimating the metrics in equations 6-9?

Maximum flow values and flood event series were not used in this study. A continuous flow series from 1990 to 2005 was utilized, where the years 1990 to 1992 were considered a "model warming-up" period (3 years). During this period, simulations were used to reduce uncertainties related to initial conditions, such as soil moisture content. The period from 1993 to 1999 was used for calibration analysis (7 years), and the period from 2000 to 2005 was used for validation analysis (6 years). The same period (1990-2005) was also employed for analyzing the projections of the climate models. The model's performance was evaluated using daily observed flows from the water level station in Três Corações, obtained from the National Water Agency (ANA).

I kindly disagree with the sentence “Nevertheless, the model (VIC) has the potential for use as a tool in modelling.” P.11 as the VIC model showed an overall not good performance. Would it be better to use it to simulate peak discharge values? Could please the Authors better explain their idea?

The authors have accepted the reviewer’s suggestion.

The time series used to calibrate the model is short, however the performance is promising. I’d kindly suggest increasing the time series length for a more robust validation.

Because the objective of the model comparison was to simulate the hydrological impact of climate change in the basin, and considering that the available scenarios correspond to the CMIP5 which baseline scenario goes to 2005, the period selected for both calibration and validation extended until 2005. We recognize however, that the series could be considered short, therefore we included a paragraph at the conclusions highlighting the need for an extended period.

Reviewer 3 Report

The manuscript evaluates the performance of three hydrological models (LASH: Lavras Simulation of Hydrology; VIC: Variable Infiltration Capacity and MHD: Distributed Hydrological Model) on a daily time step in a basin (Verde River Basin) of the of the state of Minas Gerais, Brazil.

The topic studied in the manuscript is important because hydrological models have proven to be effective tools for improving the understanding of hydrological phenomena in basins. The work is of general interest.

The authors mention (Page 11: 4. Conclusion): “The results demonstrated that LASH and MHD were able to simulate discharges with a high degree of accuracy, making them important tools for managing water resources in the basins in this study region.” Also, they state (Page 1 Abstract) “These results justify the use of LASH and MHD-INPE in studies related to discharge forecasting in the region”.

My question is:

The authors limit the interest of the LASH and MHD models only to the basins in the study area (Verde River Basin). What does this mean? These models cannot be applied to basins in other regions? Please comment.

 The manuscript is well organized and written according to the journal's instructions for authors.

 However, some improvements need to be made before publication. So, as a reviewer, I have some suggestions to improve the readability of the manuscript and also, to make it more attractive to the reader.

 Specifically,

 1. Page 1

“Consequently, the objective of this research is to assess the performance of three hydrological models (LASH, VIC, and MHD) on a daily time step in the Verde River Basin”.

should be

“The objective of this research is to assess the performance of three hydrological models (LASH, VIC, and MHD) on a daily time step in the Verde River Basin”.

COMMENT: Furthermore. I find this sentence incomplete. I believe that the authors do not adequately explain and present the reasons for evaluating the performance of these models. I suggest that they rephrase this sentence.

2. Page 4

“For this study, meteorological data were obtained…”

should be

“The  meteorological data were obtained…”

3. Page 8

“….including Viola, Mello, and Acerbi Junior (2009) and Andrade, Mello, and Beskow (2013)”

should be

“….including Viola et al. [49] and Andrade et al. [50]”

4. Page 9

Define the symbols “Q10” and “Q90”

5. Variables in the text, tables, figures and in the displayed equations should be in italic.

Minor editing of English language required

Author Response

RESOURCES

Manuscript ID: resources-2428444

Hydrological Model Performances in the Verde River Basin, Minas Gerais, Brazil

Conceição de M. M. de Oliveiraa,b ; Lívia A. Alvarengaa ; Samuel Beskowb ; Zandra Almeida da Cunhab; Marcelle Martins Vargasb ; Pâmela A. Meloa ; Javier Tomasellad ; Ana Carolina N. Santosd ; Vinicius S. O. Carvalhoe, Vinicius O. Silvaa

Answer to the Editorss

In order to adequate our work to the suggestions of the reviewers, we follow all the comments. The authors have appreciated the reviewer's valuable suggestions The answers can be observed below:

REVIEWER #3:

The authors limit the interest of the LASH and MHD models only to the basins in the study area (Verde River Basin). What does this mean? These models cannot be applied to basins in other regions? Please comment.

The Rio Verde watershed (4,100 km2) is geographically located in the southern region of Minas Gerais, within the Atlantic Forest biome, in the Serra da Mantiqueira. It is an important region for biodiversity conservation. The Serra da Mantiqueira region harbors several headwater areas that form significant rivers for water supply, irrigation, and hydropower potential in the southern part of Minas Gerais state.

These MHD and LASH models have been previously employed in other watersheds in different regions of southern Minas Gerais and other parts of Brazil, encompassing varying watershed sizes. Examples of research works utilizing the MHD hydrological model include Falck et al. (2015) in the Tocantins Araguaia River Basin, Siqueira Júnior, Tomasella, and Rodriguez (2015) in the Madeira River Basin, Rodriguez and Tomasella (2016) in the Ji-Paraná River Basin, Von Randow et al. (2019) in the Tocantins River Basin, and Melo (2020) in the Lavrinha Stream Basin.

The LASH model has also been applied in different regions, such as Viola (2015) in the Rio Grande watershed, Beskow (2016) in the Fragata River Basin, Caldeira (2019) in the Arroio Pelotas watershed, and Cunha (2021) in the Xingu River Basin.

The manuscript is well organized and written according to the journal's instructions for authors.

However, some improvements need to be made before publication. So, as a reviewer, I have some suggestions to improve the readability of the manuscript and also, to make it more attractive to the reader.

 Specifically,

  1. Page 1

“Consequently, the objective of this research is to assess the performance of three hydrological models (LASH, VIC, and MHD) on a daily time step in the Verde River Basin”.

should be:

“The objective of this research is to assess the performance of three hydrological models (LASH, VIC, and MHD) on a daily time step in the Verde River Basin”.

 COMMENT: Furthermore. I find this sentence incomplete. I believe that the authors do not adequately explain and present the reasons for evaluating the performance of these models. I suggest that they rephrase this sentence.

has been changed in the text

  1. Page 4

“For this study, meteorological data were obtained…”

should be

“The  meteorological data were obtained…”

has been changed in the text

  1. Page 8

“….including Viola, Mello, and Acerbi Junior (2009) and Andrade, Mello, and Beskow (2013)”

should be

has been changed in the text

“….including Viola et al. [49] and Andrade et al. [50]”

has been changed in the text

  1. Page 9

Define the symbols “Q10” and “Q90

has been changed in the text

  1. Variables in the text, tables, figures and in the displayed equations should be in italic.

has been changed in the text

Reviewer 4 Report

I have now revised the ms titled "Hydrological Model Performances in the Verde River Basin, Minas Gerais, Brazil". As I see, the manuscript has been already throughout a previous revision process which has helped to improve the quality of the research. All the added corrections are properly highlighted.  The research work is sound and I consider it as a good piece of science. However and prior to publication, please see below some minor comments which I hope helps to improve your research: 

Abstract: Please add future management implications for your research area taking into account your results. In addition, try to add information on the study site. It would help to better get readers' attention

Methods: Please add information about land management. Information related to forest management or agricultural land management would help better contextualize your research 

Conclusion: It would be worthy to add future implications of your work and research

Author Response

VV

Abstract: Please add future management implications for your research area taking into account your results. In addition, try to add information on the study site. It would help to better get readers' attention

Methods: Please add information about land management. Information related to forest management or agricultural land management would help better contextualize your research 

Conclusion: It would be worthy to add future implications of your work and research

We appreciate the reviewer's suggestions for improving the article. modifications were made at the indicated locations.

Round 2

Reviewer 1 Report

Authors have made the requested changes in the manuscript. The manuscript can be accepted in its present form.

Author Response

Comments and Suggestions for Authors

Authors have made the requested changes in the manuscript. The manuscript can be accepted in its present form.

We appreciate the suggestions and acceptance of the reviewer. Some improvements have been made to the article.

Reviewer 2 Report

The paper “Hydrological Model Performances in the Verde River Basin, Minas Gerais, Brazil” presents simulations and analysis performed using different hydrological models to simulate the discharge values in the Verde River Basin.

The paper presents the models and then the metrics used to estimate the goodness of fit and the parameters values used to calibrate the models. Despite the interesting topic, the models are already available from literature. 

I can see from the text that they made a substantial effort to improve the manuscript and to show the novelty. However, I'd like to point out that this looks like as an excercise and I hardly see the chances for a publication. However, it can be useful for practitioners that work in that area, it is a technical study indeed.

The text needs some minor English revision

Author Response

ENGLISH HAS BEEN REVISED 

Author Response:

R: Because these hydrological models are commonly used in studies within the Grande River Basin for predicting climate change [1 3] and land use land cover change [ 4], the uncertainties related to model structure needs to be further explored. Therefore, multi-model simulations are strategic for addressing uncertainties in runoff estimation. According to [ multi-model simulations are beneficial for identifying limitations in model structure and guide improvement strategies.
This paper not only shows the performance of three different model in simulating runoff, but states differences and limitations within simulations. For this, this study intends to reveal potential sources of uncertainties that could influence on impact prediction studies.
Goodness-of-fit statistics are commonly used during calibration process as objective functions to optimize model performance. This study has shown that only LNASH and PBIAS has indicated the poorer performance of VIC model while estimating baseflow, minimum discharge and accumulated runoff.
Therefore, for impact analysis in future studies these uncertainties related to the choice of a model or calibration method must be accounted. As this study assess variations of model performance from the same input variables, it is a reference for more reliable decisions.
1.Carvalho, V.S.O.; Alvarenga, L.A.; Oliveira, C. de M.M. de; Tomasella, J.; Colombo, A.;
Melo, P.A. Impact of Climate Change on Monthly Streamflow in the Verde River Basin
Using Two Hydrological Models. Rev. Ambient. Água 2021 , 16 , e2683, doi:10.4136/ambi
agua.2683.
2.Mello, C.R.; Vieira, N.P.A.; Guzman, J.A.; Viola, M.R.; Beskow, S.; Alvarenga, L.A.
Climate Change Impacts on Water Resources of the Largest Hydropower Plant Reservoir in
Southeast Brazil. Water 2021 , 13 , 1560, doi:10.3390/w13111560.
3.Zákhia, E.M.S.; Alvarenga, L.A.; Tomasella, J.; Martins, M.A.; Santos, A.C.N.; Melo, P.A.
Climate Change’s Impacts in a Watershed in the South of the Minas Gerais State. Rev. bras.
meteorol. 2022 , 36 , 667 681, doi:10.1590/0102 7786360002.
4.Viola, M.R.; Mello, C.R.; Beskow, S.; Norton, L.D. Impacts of Land Use Changes on the
Hydrology of the Grande River Basin Headwaters, Southeastern Brazil. Water Resources
Management 2014 , 28 , 4537 4550, doi:10.1007/s11269 014 0749 1.
5.Moges, E.; Demissie, Y.; Larsen, L.; Yassin, F. Review: Sources of Hydrological Model
Uncertainties and Advances in Their Analysis. Water 2021 , 13 , 28,
doi:10.3390/w13010028.

Round 3

Reviewer 2 Report

The last response seems convincing, the paper can be published

Minor changes should be performed